# OpenReview forum: "Self-Destructive Language Models"
_ICLR.cc/2026/Conference — ICLR 2026 Poster_

### Official Review · Reviewer_1ay1 · 2025-10-27

**Soundness:** 4
**Presentation:** 2
**Contribution:** 4
**Rating:** 8
**Confidence:** 4

**Summary:**

This work introduces a novel safety finetuning strategy that induces a strong negative coupling between gradients from adversarial examples and gradients from benign samples. The result is a model that largely preserves the harmlessness and helpfulness of the base model and is such that, when adversarial finetuning aimed at stripping safety guardrails is applied, either the model remains safe, or the model is so degraded as to become useless. This is achieved by adding a term to the loss that promotes dissimilarity between gradients from contrastive sets. A hessian-free estimate is proposed for the gradient of this loss to make it practical.

**Strengths:**

* The proposed method is clever, non-obvious, and proved effective
* The Hessian-free gradient estimate for $\mathcal{L}_{SD}$ is a valuable contribution in itself and is likely to find further applications
* The paper is clearly written and sound.
* The experiments are sufficiently complete and support the claims.
* The problem is highly relevant and pressing, especially for the developers of open-weight models.

**Weaknesses:**

* [W1] There is evidence (e.g in the cited [Qi et al. 2023]) that also well-intended fine-tuning leads to catastrophic forgetting of safeguards. The paper does not consider this aspect. While the contributions in the paper are already sufficient for making it worth including at ICLR, it may create a blind spot and a false sense of security unless either there are experiments on the effects of non-adversarial fine-tuning, or this limitations is emphasized upfront.

**Questions:**

* [Q1] What would be the effect of applying to a SEAM model the weight orthogonalization procedure of [Arditi et al. 2024]? Since that method is not based on gradient descent, could it still be effective?
* [Q2] It would be interesting to see a plot of $\mathcal{L}_{SD}$: is it driven all the way down to -1 (in the case of cosine similarity)? How rapidly?

Suggestions and comments:

L031-L33: garbled citation text, Latex error.

L157-161: At this point this is redundant.

L195-197: What is exactly the logarithmic transformation, and how is it involved in preventing catastrophic forgetting?

Theorem 1: how tight is the bound? How does one estimate the Lipschitz constant?

L275-276: $\beta$ is small. What are the variances of the different loss components? They would be useful to understand the real relative importance of the loss terms.

L362-364: There is no model in Figure 3(b) for which the harmfulness may be said to be 'high'.

L378-393: Layout too compressed

[Arditi et al. 2024] Arditi, A., Obeso, O., Syed, A., Paleka, D., Panickssery, N., Gurnee, W. and Nanda, N., 2024. Refusal in language models is mediated by a single direction. Advances in Neural Information Processing Systems, 37, pp.136037-136083.

---

> ### Author Response · Authors · 2025-11-20
>
> We thank the reviewer for the valuable feedback on improving this paper! Please find below our response to the reviewer’s questions.
>
> ---
>
> > There is evidence (e.g in the cited [Qi et al. 2023]) that also well-intended fine-tuning leads to catastrophic forgetting of safeguards. The paper does not consider this aspect. While the contributions in the paper are already sufficient for making it worth including at ICLR, it may create a blind spot and a false sense of security unless either there are experiments on the effects of non-adversarial fine-tuning, or this limitations is emphasized upfront.
>
> We appreciate the reviewer's question. We tested fine-tuning on both a pure benign dataset (Alpaca) and a benign dataset polluted with a small fraction of harmful data. The results in Table 2 demonstrate that the harmfulness score (HS) and zero-shot score (ZS) of SEAM-defended models remain stable after fine-tuning under these settings, indicating that such models are neither jailbroken nor degraded by benign data.
>
> Further, we conduct fine-tuning experiments using additional datasets (details in Section 5.4 of the revised paper). We fine-tune base and SEAM-protected models on the OR-Bench benchmark [1], which comprises sensitive yet benign prompts; we randomly sample 2,000 prompts paired with GPT-5-mini-generated responses and apply LoRA-based supervised fine-tuning for 3 epochs. In addition to their zero-shot performance (ZS) on the EleutherAI's LM Evaluation Harness benchmark, we further evaluate their over-refusal behavior on the XSTest dataset [2] using two metrics: (i) Incorrect Refusal Rate (IRR), measuring refusals on safe prompts, and (ii) Correct Refusal Rate (CRR), measuring refusals on unsafe prompts.
>
> The table below summarizes the IRR, CRR, and ZS of base and SEAM-protected models before and after fine-tuning. First, SEAM exhibits a lower IRR than the base model prior to fine-tuning, indicating its lower over-refusal behavior. Second, SEAM's IRR further decreases after fine-tuning while its CRR remains high, indicating that the model is not compromised by fine-tuning and maintains robust safety guardrails. Finally, SEAM's ZS remains stable throughout fine-tuning, confirming that fine-tuning on sensitive yet benign data does not induce catastrophic forgetting or degrade model capabilities.
>
> |          |  IRR |  CRR |  ZS  |
> |----------|:----:|:----:|:----:|
> | Base     | 0.24 | 1.00 | 51.6 |
> | SEAM     | 0.16 | 1.00 | 50.8 |
> | Base (after fine-tuning)  | 0.14 | 0.90 | 52.6 |
> | SEAM (after fine-tuning) | 0.08 | 0.98 | 52.4 |
>
> ---
>
> > What would be the effect of applying to a SEAM model the weight orthogonalization procedure of [Arditi et al. 2024]? Since that method is not based on gradient descent, could it still be effective?
>
> Following the reviewer's suggestion, we implement the weight orthogonalization attack using the paper's default settings for hyperparameters, datasets, and metrics, and report the attack success rate (ASR) in the table below (details in Appendix C.5). Notably, both the base Llama-2-7B and SEAM-defended models can be effectively attacked by this orthogonalization attack. To mitigate this vulnerability, we identify Extended-Refusal (ER) [3] as a potential defense, which fine-tunes models using a dataset of responses to harmful prompts that provide detailed justifications before refusing, thereby distributing the refusal signal across multiple token positions. We combine SEAM with ER (SEAM-ER) by fine-tuning SEAM-defended models on 500 examples containing more diverse refusal responses generated by GPT-5-mini to obscure the refusal direction. This simple augmentation substantially reduces ASR. These results suggest that combining SEAM with the fine-grained dataset and pipeline from [3] can help build more comprehensive defenses.
>
> |      | Base | SEAM | SEAM-ER |
> |:---:|:------------:|:------------:|:------------------:|
> | ASR |    0.99     |     0.98     |        0.36        |
>
> ---
>
> > It would be interesting to see a plot of is it driven all the way down to -1 (in the case of cosine similarity)? How rapidly?
>
> We appreciate the reviewer’s question. Figure 8 in Appendix C.11 presents different loss components during training. The loss $L_{sd}$ quickly decreases to approximately -1 within about 250 steps, and subsequently stabilize. This dynamic pattern indicates that the objective is being effectively optimized. We also present the dynamics of other loss terms (see Figure 8 for details), showing effective optimization convergence.
>
> ---
>
> [1] Cui et al., OR-Bench: An Over-Refusal Benchmark for Large Language Models, ICML’25\
> [2] Röttger et al., XSTest: A test suite for identifying exaggerated safety behaviours in large language models. NAACL’24.\
> [3] Shairah et al., An Embarrassingly Simple Defense Against LLM Abliteration Attacks, 2025

---

> ### Author Response · Authors · 2025-11-20
>
> > L031-L33: garbled citation text, Latex error.
>
> > L157-161: At this point this is redundant.
>
> > L378-393: Layout too compressed
>
> Following the reviewer’s suggestion, we have revised the presentation accordingly.
>
> ---
>
> > L195-197: What is exactly the logarithmic transformation, and how is it involved in preventing catastrophic forgetting?
>
> Instead of using the loss term $L_{sd}$ directly, we optimize its logarithm $\log⁡(L_{sd})$. This technique is also employed in prior work (e.g., RepNoise). Intuitively, it reduces the magnitude of gradient updates when the loss becomes large, helping to prevent catastrophic forgetting, particularly important since gradient ascent inherently increases the loss.
>
> ---
>
> > Theorem 1: how tight is the bound? How does one estimate the Lipschitz constant?
>
> The bound in Theorem 1 is an upper bound derived from a second-order Taylor expansion. Theoretically, the error is dominated by the linear term $\epsilon$, meaning the bound is tight up to $\mathcal{O}(\epsilon)$. Empirically, our ablation study on perturbation magnitude (Figure 5) aligns with the theoretical prediction that the error scales with $\epsilon$.
>
> Moreover, directly estimating the local Hessian Lipschitz constant ($L^H$) for LLMs is computationally intractable as it requires extensive Hessian computations. In practice, we do not estimate $L^H$ explicitly. Instead, we treat $\epsilon$ as a tunable hyperparameter. As noted in our analysis, we select $\epsilon$ to be small enough to minimize the approximation error (relative to the implicit $L^H$) but large enough to avoid numerical instability. Our empirical results indicate that $\epsilon=1e-3$ provides the optimal balance.
>
> ---
>
> > \beta is small. What are the variances of the different loss components? They would be useful to understand the real relative importance of the loss terms.
>
> We appreciate the reviewer's question. In Eq. (5), the weight of $L_{ul}$​ (gradient ascent loss) is 1 and the weight of $L_{sd}$​ (self-destructive loss) is $\beta$, so we examine the variances of these two core loss components to assess whether the choice of $\beta$ is reasonable. We compute the batch-level variance of both losses at the beginning of training, after 200 steps, and after 400 steps, and derive $\beta$ by computing their variance ratio. As shown in the table below, the ideal $\beta$ is on the order of 1e-2, which is consistent with our empirical setting of $\beta = 0.01$. Additionally, the training dynamics of $L_{sd}$ in Appendix C.11 show meaningful decreases, indicating that although $\beta$ is small, it still contributes effectively to the overall optimization.
>
>
>
> |                              | Beginning | 200 Steps | 400 Steps |
> |------------------------------|:---------:|:---------:|:---------:|
> | Variance of $L_{ul}$         | 3.06×1e-4 | 6.07×1e-4 | 1.31×1e-3 |
> | Variance of $L_{sd}$         | 1.31×1e-2 | 9.72×1e-3 | 7.65×1e-3 |
> | Variance Ratio/Ideal $\beta$ |   0.023   |   0.062   |    0.17   |
>
> ---
>
> Again, we thank the reviewer for the valuable feedback. Please let us know if there are any other questions or suggestions.
>
> Best,
>
> Authors

---

### Official Review · Reviewer_b8Na · 2025-10-31

**Soundness:** 3
**Presentation:** 3
**Contribution:** 3
**Rating:** 6
**Confidence:** 4

**Summary:**

This paper addresses the vulnerability of Large Language Models (LLMs) to harmful fine-tuning attacks. The authors state that existing defenses do not sufficiently address the models' inherent 'trainability' on harmful data.

The primary contribution is SEAM, a novel alignment defense that transforms an LLM into a "self-destructive model". This model is designed to retain its performance on legitimate (benign) tasks while exhibiting substantial performance degradation when an adversary attempts to fine-tune it on harmful data.

This self-destructive mechanism is achieved through a new loss function that couples the optimization trajectories of benign and harmful data. Specifically, it encourages their respective gradients to point in opposing directions. This effect is amplified using adversarial gradient ascent.

To make this optimization computationally practical, the authors also develop an efficient Hessian-free gradient estimate and provide its theoretical error bounds.

Evaluations demonstrate that SEAM models show robustness against low-intensity attacks and undergo significant performance collapse, rendering them unusable, under high-intensity attacks.

**Strengths:**

Originality: The paper's primary strength is its originality. The proposal to counter harmful fine-tuning by making the model "self-destructive" is a technically novel concept. This approach moves beyond simply reinforcing existing alignment and instead introduces a mechanism that actively degrades the model's core capabilities when a harmful objective is pursued. This formulation presents a new direction for model defense.

Quality: The paper exhibits high quality in its technical execution. The theoretical background for the proposed self-destructive loss function is well-presented. The authors provide a practical Hessian-free gradient estimation to address the computational challenges of their method, complete with theoretical error bounds, which demonstrates technical depth. The empirical evaluation is extensive, testing the defense against various attack intensities and methods (SFT, PEFT) and including relevant ablation studies to validate the components of the loss function.

Clarity: The paper is clearly written. The problem of harmful fine-tuning and the limitations of existing defenses are well-defined. The proposed solution, SEAM, is explained in a structured manner, and the core concept of self-destruction is illustrated effectively (e.g., in Figure 1).

Significance: The work is significant as it addresses a critical and persistent vulnerability in large language models. By proposing a defense that creates a "no-win situation" for adversaries—where attacks either fail to compromise safety or destroy the model's general utility—the paper offers a promising alternative to current alignment techniques that remain vulnerable to more intensive attacks.

**Weaknesses:**

A primary weakness concerns the practical applicability of the method, particularly regarding hyperparameter tuning. The paper relies on grid search to find optimal values for key hyperparameters such as alpha, beta, and epsilon. This approach was feasible for the experimental setup, which used relatively small-scale models (in the 3B to 8B parameter range) and datasets. However, this tuning method does not scale efficiently to large, production-grade foundation models, where the computational cost of repeated grid search would be prohibitive. The paper would be significantly strengthened by providing a more detailed sensitivity analysis for these parameters or, ideally, discussing a more principled or efficient method for setting them, which would be crucial for real-world adoption.

Furthermore, the main evaluation framework for attack intensity is a limitation. The study largely equates "low-intensity" and "high-intensity" attacks with the learning rate used during the harmful fine-tuning process. This assumption about adversarial strategy is somewhat naive. A sophisticated adversary has more variables to control. For instance, an attacker could use a low or moderate learning rate over a significantly longer number of training steps to gradually compromise the model. This might allow them to find a path that bypasses the sharp self-destruction mechanism that the paper demonstrates is triggered by high learning rates. The current evaluation does not fully explore this trade-off between learning rate and the duration of the attack.

**Questions:**

1.  The paper mentions that grid search was used to find the optimal hyperparameters (e.g., alpha, beta, epsilon). For clarity and to aid in reproducibility, it would be helpful if the authors could provide the specific ranges and values tested for each hyperparameter during this grid search.

2.  The default harmful fine-tuning attack is set at 1K harmful samples and 250 training steps. Could the authors provide more justification for this setting as an adequate representation of a typical attack? We are particularly interested in the authors' perspective on what might happen if the attack was more intensive in terms of data volume or duration, rather than just learning rate. For example, what effect would be expected if an adversary used 10K samples and trained for 2,500 steps? Would the self-destructive mechanism still be triggered as effectively?

---

> ### Author Response · Authors · 2025-11-20
>
> We thank the reviewer for the valuable feedback on improving this paper! Please find below our response to the reviewer’s questions.
>
> ---
>
> > A primary weakness concerns the practical applicability of the method, particularly regarding hyperparameter tuning. The paper relies on grid search to find optimal values for key hyperparameters such as alpha, beta, and epsilon. This approach was feasible for the experimental setup, which used relatively small-scale models (in the 3B to 8B parameter range) and datasets. However, this tuning method does not scale efficiently to large, production-grade foundation models, where the computational cost of repeated grid search would be prohibitive. The paper would be significantly strengthened by providing a more detailed sensitivity analysis for these parameters or, ideally, discussing a more principled or efficient method for setting them, which would be crucial for real-world adoption.
>
> > The paper mentions that grid search was used to find the optimal hyperparameters (e.g., alpha, beta, epsilon). For clarity and to aid in reproducibility, it would be helpful if the authors could provide the specific ranges and values tested for each hyperparameter during this grid search.
>
> We appreciate the reviewer’s question. We have conducted a sensitivity analysis of SEAM's hyperparameters $\alpha$, $\beta$, and $\beta$ in Tables 16 and 17 (Appendix C.7) and Figure 5. The tested ranges are:  $\alpha \in [0.01, 0.1, 0.5, 1, 5]$, $\beta \in [1e-4, 0.001, 0.01, 0.1, 1]$, and $\epsilon \in  [1e-6, 1e-5, 1e-4, 1e-3, 1e-2, 1e-1]$. Moreover, we apply this parameter setting across various models in the experiments of Table 6, indicating that these relative loss weights generalize well. Additionally, since our focus is on safeguarding open-source models, which are often of moderate sizes, a limited grid search is both practical and sufficient.
>
> ---
>
> > Furthermore, the main evaluation framework for attack intensity is a limitation. The study largely equates "low-intensity" and "high-intensity" attacks with the learning rate used during the harmful fine-tuning process. This assumption about adversarial strategy is somewhat naive. A sophisticated adversary has more variables to control. For instance, an attacker could use a low or moderate learning rate over a significantly longer number of training steps to gradually compromise the model. This might allow them to find a path that bypasses the sharp self-destruction mechanism that the paper demonstrates is triggered by high learning rates. The current evaluation does not fully explore this trade-off between learning rate and the duration of the attack.
>
> > The default harmful fine-tuning attack is set at 1K harmful samples and 250 training steps. Could the authors provide more justification for this setting as an adequate representation of a typical attack? We are particularly interested in the authors' perspective on what might happen if the attack was more intensive in terms of data volume or duration, rather than just learning rate. For example, what effect would be expected if an adversary used 10K samples and trained for 2,500 steps? Would the self-destructive mechanism still be triggered as effectively?
>
> We appreciate the reviewer’s question. We set 1K samples and 250 training steps as the default setting to enable comparison with prior work (e.g., RepNoise). In Figure 3, we explore sample sizes ranging from 1K to 10K. Intuitively, with 10K samples and 250 steps, a learning rate of 1e-4 (Attack #7) causes complete utility collapse, while a learning rate of 5e-5 (Attack #6) does not cause significant utility drop but also fails to increase the harmfulness score.
>
> Additionally, we consider attacks with extremely low learning rates and significantly more steps (from 2.5K to 50K steps) using 10K samples. The results are shown in the table below. The SEAM is resistant to this  low-intensity-long-duration attack as its gradient direction can still trigger the gradient trap. Moreover, the magnitude of self-destruction gracefully scales with the degree of alignment compromise, as the increasing attack intensity (learning rates) leads to progressively greater utility degradation (more details in Appendix C.5).
>
>
> |                            | HS  | ZS   |
> |----------------------------|-----|------|
> | $\eta$ = 5e-6, 2.5k steps  | 9.7 | 40.5 |
> | $\eta$ = 1e-6, 2.5k steps  | 6.3 | 45.5 |
> | $\eta$ = 5e-6, 12.5k steps | 0.0 | 25.7 |
> | $\eta$ = 1e-6, 12.5k steps | 9.9 | 39.2 |
> | $\eta$ = 1e-6, 50k steps | 0.0 | 25.7 |
> | $\eta$ = 5e-6, 50k steps | 0.0 | 25.0 |
>
> ---
>
> Again, we thank the reviewer for the valuable feedback. Please let us know if there are any other questions or suggestions.
>
> Best,
>
> Authors

---

### Official Review · Reviewer_tToc · 2025-10-31

**Soundness:** 3
**Presentation:** 2
**Contribution:** 3
**Rating:** 6
**Confidence:** 3

**Summary:**

This paper presents SEAM, an alignment-enhancing defense against harmful fine-tuning. Unlike prior defenses, SEAM introduces a loss function that couples the optimization trajectories of benign and harmful data. Its main contribution is an efficient, Hessian-free gradient-based optimization with a theoretical guarantee.

**Strengths:**

1. This paper introduces SEAM, which creates an optimization trap by coupling benign and harmful optimization trajectories
2. The paper validates SEAM's effectiveness through extensive evaluation across a diverse range of LLMs and various attack configurations, demonstrating SEAM's robust resistance to low-intensity attacks and self-destruction under high-intensity attacks
3. The mechanism is validated through PCA visualization of gradients, confirming that SEAM successfully forces benign and adversarial gradients into opposite directions. The theoretical proof is also provided.

**Weaknesses:**

1.  The success of the gradient coupling mechanism critically depends on the adversarial dataset and the benign dataset, and to ensure consistently opposing gradients. Therefore, the robustness highly depends on the quality and representativeness of these datasets, rather than their quantity (Fig.3). As the authors also knowledge, there is a need to "explore identifying or generating optimal benign datasets that maximize the self-destructive effect".
2. The core success of SEAM is turning a strong harmful fine-tuning attack (aiming for a jailbroken model) into model collapse (aiming for destruction). However, if the attacker's primary objective is to disrupt model availability or utility (not jailbreak), the self-destruction mechanism, which causes performance drop, actually helps in achieving this attacking goal. This suggests the defense effectively shifts the attack outcome rather than neutralizing all adversarial objectives.
3. The defense method against attacks incorporating benign data mixing, benign task regularizers, and random gradient perturbations is tested. However, other strategies, such as fine-tuning only non-critical layers, using a covert target/loss to avoid the self-destruction trigger condition, are not extensively evaluated.
4. Some writing and presentation issues. For example, there is a missing full stop at the end of the first paragraph, and some references appear incorrectly formatted in the first sentence.

**Questions:**

1. How does SEAM perform under stronger or more adaptive attack settings (such as training with extremely low learning rates or fine-tuning only specific LoRA layers)? Can the authors provide empirical evidence that SEAM remains effective in these challenging cases?
2. More detailed analysis of the attacking samples is needed, not only in terms of their quantity, but also their characteristics compared to normal samples. For example, examining their distributional differences could provide deeper insight into how SEAM responds to harmful data.

---

> ### Author Response · Authors · 2025-11-20
>
> We thank the reviewer for the valuable feedback on improving this paper! Please find below our response to the reviewer’s questions.
>
> ---
>
> > The success of the gradient coupling mechanism critically depends on the adversarial dataset and the benign dataset, and to ensure consistently opposing gradients. Therefore, the robustness highly depends on the quality and representativeness of these datasets, rather than their quantity (Fig.3). As the authors also knowledge, there is a need to "explore identifying or generating optimal benign datasets that maximize the self-destructive effect".
>
> We thank the reviewer’s question. We argue that while some benign datasets may be more informative, many datasets can provide sufficiently representative gradient information for modeling utility improvement/degradation. To verify this, we conduct additional experiments using alternative datasets other than the Alpaca dataset. Specifically, we use MathQA, a domain-specific dataset that differs substantially from the general Alpaca dataset. We randomly select 4,000 samples to construct the benign dataset (more details in Appendix C.10 in the revised paper). The table below reports the harmfulness score (HS) and the zero-shot score (ZS) results under attacks with varying learning rates $\eta$, demonstrating that the choice of benign dataset is not critical.
>
>
> |      | Pre-attack |      | $\eta$ = 2e-5 |      | 5e-5 |      | 8e-5 |      | 1e-4 |      | 2e-4 |      |
> |------|:----------:|:----:|:-------------:|:----:|:----:|:----:|:----:|:----:|:----:|:----:|:----:|:----:|
> |      |     HS     |  ZS  |       HS      |  ZS  |  HS  |  ZS  |  HS  |  ZS  |  HS  |  ZS  |  HS  |  ZS  |
> | Base |     5.0    | 51.6 |      47.3     | 51.7 | 77.5 | 50.6 | 80.4 | 49.7 | 78.8 | 49.8 | 79.5 | 50.2 |
> | SEAM |     5.0    | 50.8 |      4.2      | 47.1 |  9.4 | 40.4 |  5.1 | 37.4 |  0.0 | 26.3 |  0.0 | 25.4 |
>
> ---
>
> > The core success of SEAM is turning a strong harmful fine-tuning attack (aiming for a jailbroken model) into model collapse (aiming for destruction). However, if the attacker's primary objective is to disrupt model availability or utility (not jailbreak), the self-destruction mechanism, which causes performance drop, actually helps in achieving this attacking goal. This suggests the defense effectively shifts the attack outcome rather than neutralizing all adversarial objectives.
>
> We agree with the reviewer that availability attacks represent a major threat to proprietary LLMs accessed through APIs (e.g., disrupting API services). However, in this paper, as discussed in Section 3, we assume a white-box setting where the attacker maliciously fine-tunes an open-source model and releases the jailbroken model for misuse. Under this threat model, if the fine-tuned model collapses and its utility degrades, the attacker's released model becomes useless, making SEAM's self-destruction outcome a successful defense. Note that under this threat model, the adversary has full control over the fine-tuning data (without concern for filtering) and can precisely calibrate attack parameters (e.g., learning rate and optimizer), thereby representing a stronger threat than API-based attacks.

---

> ### Author Response · Authors · 2025-11-20
>
> > The defense method against attacks incorporating benign data mixing, benign task regularizers, and random gradient perturbations is tested. However, other strategies, such as fine-tuning only non-critical layers, using a covert target/loss to avoid the self-destruction trigger condition, are not extensively evaluated.
>
> Following the reviewer's suggestion, we consider fine-tuning only non-critical layers by freezing the intermediate layers (layers 10–20) of Llama 2, which are often regarded as critical layers. As shown in the table below, this attack weakens the utility degradation while the harmfulness score remains low. The likely explanation is that updating non-critical layers is less effective for harmful fine-tuning. Additionally, since SEAM applies gradient trapping across all layers, the self-destruction effect persists even when critical layers are frozen (details in Appendix C.5 of the revised paper).
>
> |      | Pre-attack |      | $\eta$ = 2e-5 |      | 5e-5 |      | 8e-5 |      | 1e-4 |      | 2e-4 |      |
> |------|:----------:|:----:|:-------------:|:----:|:----:|:----:|:----:|:----:|:----:|:----:|:----:|:----:|
> |      |     HS     |  ZS  |       HS      |  ZS  |  HS  |  ZS  |  HS  |  ZS  |  HS  |  ZS  |  HS  |  ZS  |
> | Base |     5.0    | 51.6 |      47.3     | 51.7 | 77.5 | 50.6 | 80.4 | 49.7 | 78.8 | 49.8 | 79.5 | 50.2 |
> | SEAM |     5.0    | 50.8 |      5.0      | 49.3 |  6.1 | 45.1 |  6.7 | 42.5 |  6.4 | 37.3 |  0.0 | 25.4 |
>
> Moreover, we implement a reversal attack that augments the adversary's loss function with the $L_{\text{sd}}$​ loss term (using $\beta = -0.01$), thereby simultaneously pursuing the adversarial objective while attempting to escape the gradient trap (see Appendix C.5 for details). The table below reports post-attack HS and ZS across different learning rates $\eta$, showing that SEAM remains robust against such adaptive attacks. We attribute this resilience to the difficulty of retracing the optimization trajectory once the model has converged to a gradient-trap basin: the highly non-convex loss landscape makes it challenging to reverse the path by simply maximizing $L_{\text{sd}}$​ from that local region.
>
> | $\eta$ = 2e-5 |      | 5e-5 |      | 8e-5 |      | 1e-4 |      | 2e-4 |      |
> |:--------:|:----:|:----:|:----:|:----:|:----:|:----:|:----:|:----:|:----:|
> |    HS    |  ZS  |  HS  |  ZS  |  HS  |  ZS  |  HS  |  ZS  |  HS  |  ZS  |
> |    5.3   | 49.3 |  8.2 | 35.5 |  0.0 | 25.6 |  0.0 | 25.1 |  0.0 | 26.2 |
>
> ---
>
> > Some writing and presentation issues. For example, there is a missing full stop at the end of the first paragraph, and some references appear incorrectly formatted in the first sentence.
>
>
> We appreciate the reviewer for pointing this out. We have revised the manuscript accordingly.
>
> ---
>
> > How does SEAM perform under stronger or more adaptive attack settings (such as training with extremely low learning rates or fine-tuning only specific LoRA layers)? Can the authors provide empirical evidence that SEAM remains effective in these challenging cases?
>
> Following the reviewer’s suggestion, we consider attacks with extremely low learning rates and significantly more steps (from 2.5K to 50K steps) using 10K samples. The results are shown in the table below. The SEAM is resistant to this  low-intensity-long-duration attack as its gradient direction can still trigger the gradient trap. Moreover, the magnitude of self-destruction gracefully scales with the degree of alignment compromise, as the increasing attack intensity (learning rates) leads to progressively greater utility degradation (more details in Appendix C.5).
>
>
> |                            | HS  | ZS   |
> |----------------------------|-----|------|
> | $\eta$ = 5e-6, 2.5k steps  | 9.7 | 40.5 |
> | $\eta$ = 1e-6, 2.5k steps  | 6.3 | 45.5 |
> | $\eta$ = 5e-6, 12.5k steps | 0.0 | 25.7 |
> | $\eta$ = 1e-6, 12.5k steps | 9.9 | 39.2 |
> | $\eta$ = 1e-6, 50k steps | 0.0 | 25.7 |
> | $\eta$ = 5e-6, 50k steps | 0.0 | 25.0 |
>
>
> Further, in addition to applying LoRA on "q_proj" and "v_proj" in Attack #8 and Attack #9 in Figure 3, we further consider applying LoRA only on "q_proj", "k_proj", "v_proj", "o_proj", "gate_proj", "up_proj", "down_proj". The table below shows that involving more parameters leads to greater ZS drop, indicating the effectiveness of the self-destructive effect (see Appendix C.5 for details).
>
> |    | Attack #8 | Attack #9 | Attack #8  Full Modules | Attack #9  Full Modules |
> |----|:---------:|:---------:|:-----------------------:|:-----------------------:|
> | HS |    5.07   |    6.36   |           8.6           |           7.6           |
> | ZS |    48.6   |    47.7   |           45.3          |           43.6          |

---

> ### Author Response · Authors · 2025-11-20
>
> > More detailed analysis of the attacking samples is needed, not only in terms of their quantity, but also their characteristics compared to normal samples. For example, examining their distributional differences could provide deeper insight into how SEAM responds to harmful data.
>
> We analyze the gradients of harmful and benign data on SEAM in Figure 6, showing that they exhibt distinct patterns under a PCA projection. We further present numerical measures to quantify these distinctions. The blue cluster corresponds to gradients from 100 randomly sampled benign batches, with an average norm of 532.11 and a mean distance to its centroid of 5.69 (indicating tight clustering). In contrast, the red cluster corresponds to 100 randomly sampled harmful batches, exhibiting an average norm of 984.50 and a mean distance to its centroid of 699.10. Moreover, the average cosine similarity between benign and harmful gradient pairs is approximately −0.703, indicating nearly opposite directions. These measures suggest that SEAM induces substantial distributional shifts between harmful and benign gradients, fulfilling its design objectives in Section 4.1. We have revised the paper to incorporate this analysis.
>
> ---
>
> Again, we thank the reviewer for the valuable feedback. Please let us know if there are any other questions or suggestions.
>
> Best,
>
> Authors

---

### Official Review · Reviewer_Y52o · 2025-11-01

**Soundness:** 3
**Presentation:** 3
**Contribution:** 3
**Rating:** 6
**Confidence:** 4

**Summary:**

This paper proposes a defence method against harmful fine-tuning attacks. The proposed method SEAM puts the model in a peculiar zone in the parameter space where any attempt of harmful fine-tuning will result in partial or total degradation of the model’s benign capabilities, therefore making the attack useless. The method achieves that by optimizing a self-distruction loss which tries to minimize the alignment between the potential gradients of harmful and benign fine-tuning. The experimental results show that the method has superior defence capabilities compared to previous works when evaluated on a wide range of harmful fine-tuning settings.

**Strengths:**

The paper approaches an interesting and practical issue: harmful fine-tuning of LLMs. The proposed solution SEAM solves this problem in a clever and original way: by self-destructing the model’s capabilities when harmful fine-tuning is attempted.

The paper is well structured and easy to read. The main idea is clearly explained and the presentation intuitively builds the solution.

The Hessian-free gradient estimate makes the optimization problem tractable and the method’s costs comparable to other related methods tackling the same problem.

The paper presents thorough experiments on multiple settings and extensive comparisons with related works. The additional experiments, ablations, visualizations and insights are also interesting and valuable.

**Weaknesses:**

**Limited fine-tuning evidence**

The paper claims that SEAM does not interfere with a model's ability to be fine-tuned on benign datasets, but there’s not a lot of evidence in this regard. As I understand, Table 1 reports model performance after fine-tuning on 4 tasks in the FS column, but these tasks are not very diverse and are likely aligned with the model’s base training, so they might not be very challenging to begin with. I would like to see some experiments with fine-tuning on more diverse / OOD benign datasets. (See also Question 2.)

I also feel that the structure of Table 1 is not obvious on the first read, maybe the caption should be a bit more explicit. I think you should add at least something like: “... zero-shot (ZS) and fine-tuning (FS) capabilities …”, but a more detailed caption or even a separate table if you decide to add some more datasets would be beneficial.

**Limited Utility Results**

Table 1 presents accuracy results for SEAM-trained models on MCQ tasks. While useful for evaluating any performance degradation, MCQ tasks are rather limited. For a more thorough evaluation, the paper should include results on some open-ended text generation tasks, like MT-Bench [1].

I would also be interested to see the model’s results on XSTest [2] - a dataset specifically designed to assess the models’ tendency to be overly cautious when answering benign questions formulated to closely resemble harmful ones. While SEAM does not directly do refusal training, I would like to see if it interferes with the model's ability to properly answer XSTest questions.

**Difficulties to implement in practice**

While the idea is interesting and effective, and the costs might be negligible compared to pre-training, it is very unlikely that any open-source models will be released with SEAM safe-guards, because companies would not risk their models collapsing when users try to fine-tune them. While I don’t see it as a weakness of the paper, I would like to hear the authors’ opinion on this issue.

**Minor**

The first sentence seems to have a wrong \citep command which does not render the citations properly.

**References**

[1] https://arxiv.org/abs/2306.05685

[2] https://arxiv.org/abs/2308.01263

**Questions:**

1. Would it be possible to undo the SEAM training by training to maximize the self-destruction loss (essentially training with negative beta)?
2. Could SEAM trigger self-destruction when finetuned on some out-of-distribution domain-specific datasets that might come around as potentially not safe? I am thinking of medical datasets where medical advice could be classified by some models as harmful, or science datasets which could contain descriptions of dangerous chemicals / details about how atomic bombs work etc.
3. Is it possible that SEAM-trained models might be vulnerable to poison attacks? I think that including a few harmful (or even specifically crafted) examples in a benign dataset could cause the model to collapse. Would it be possible to actually identify such samples from their gradient information?

---

> ### Author Response · Authors · 2025-11-20
>
> We thank the reviewer for the valuable feedback on improving this paper! Please find below our response to the reviewer’s questions.
>
> ---
>
> > Limited fine-tuning evidence: The paper claims that SEAM does not interfere with a model's ability to be fine-tuned on benign datasets, but there’s not a lot of evidence in this regard. As I understand, Table 1 reports model performance after fine-tuning on 4 tasks in the FS column, but these tasks are not very diverse and are likely aligned with the model’s base training, so they might not be very challenging to begin with. I would like to see some experiments with fine-tuning on more diverse / OOD benign datasets. (See also Question 2.)
>
> We appreciate the reviewer for the valuable suggestions. To answer this question, we conduct experiments using additional datasets (details in Section 5.4 of the revised paper). We fine-tune base and SEAM-defended models on the ORBench benchmark [1], which comprises sensitive yet benign prompts; we randomly sample 2,000 prompts paired with GPT-5-mini-generated responses and apply LoRA-based supervised fine-tuning for 3 epochs. In addition to their zero-shot performance (ZS) on the EleutherAI's LM Evaluation Harness benchmark, we further evaluate their over-refusal behavior on the XSTest dataset [2] using two metrics: (i) Incorrect Refusal Rate (IRR), measuring refusals on safe prompts, and (ii) Correct Refusal Rate (CRR), measuring refusals on unsafe prompts.
>
> The table below summarizes the IRR, CRR, and ZS of base and SEAM-defended models before and after fine-tuning. First, SEAM exhibits a lower IRR than the base model prior to fine-tuning, indicating its lower over-refusal behavior. Second, SEAM's IRR further decreases after fine-tuning while its CRR remains high, indicating that the model is not compromised by fine-tuning and maintains robust safety guardrails. Finally, SEAM's ZS remains stable throughout fine-tuning, confirming that fine-tuning on sensitive yet benign data does not induce catastrophic forgetting or degrade model capabilities.
>
> |          |  IRR |  CRR |  ZS  |
> |----------|:----:|:----:|:----:|
> | Base     | 0.24 | 1.00 | 51.6 |
> | SEAM     | 0.16 | 1.00 | 50.8 |
> | Base (after fine-tuning)  | 0.14 | 0.90 | 52.6 |
> | SEAM (after fine-tuning) | 0.08 | 0.98 | 52.4 |
>
> ---
>
> > I also feel that the structure of Table 1 is not obvious on the first read, maybe the caption should be a bit more explicit. I think you should add at least something like: “... zero-shot (ZS) and fine-tuning (FS) capabilities …”, but a more detailed caption or even a separate table if you decide to add some more datasets would be beneficial.
>
> Following the reviewer’s suggestion, we have revised Table 1’s caption.
>
> ---
>
> > Table 1 presents accuracy results for SEAM-trained models on MCQ tasks. While useful for evaluating any performance degradation, MCQ tasks are rather limited. For a more thorough evaluation, the paper should include results on some open-ended text generation tasks, like MT-Bench [1]. I would also be interested to see the model’s results on XSTest [2] - a dataset specifically designed to assess the models’ tendency to be overly cautious when answering benign questions formulated to closely resemble harmful ones. While SEAM does not directly do refusal training, I would like to see if it interferes with the model's ability to properly answer XSTest questions.
>
> Following the reviewer’s suggestion, we use the FastChat implementation [3] to evaluate the average utility of base and SEAM-trained models on MT-bench. The table below shows that both models achieve similar utility scores. We further test SEAM’s incorrect refusal rate (IRR) on XSTest (details in Section 5.4 of the revised paper). Notably, compared with the base model, SEAM is less likely to refuse sensitive but safe prompts.
>
> |          |  Utility (MT-Bech) | IRR (XSTest) |
> |----------|:----:|:----:|
> | Base     | 6.26 | 0.24 |
> | SEAM     | 5.93 | 0.16 |
>
> ---
>
> [1] Cui et al., OR-Bench: An Over-Refusal Benchmark for Large Language Models, ICML’25\
> [2] Röttger et al., XSTest: A test suite for identifying exaggerated safety behaviours in large language models. NAACL’24.\
> [3] FastChat. https://github.com/lm-sys/FastChat.

---

> ### Author Response · Authors · 2025-11-20
>
> > Difficulties to implement in practice: While the idea is interesting and effective, and the costs might be negligible compared to pre-training, it is very unlikely that any open-source models will be released with SEAM safe-guards, because companies would not risk their models collapsing when users try to fine-tune them. While I don’t see it as a weakness of the paper, I would like to hear the authors’ opinion on this issue.
>
> We thank the reviewer for raising this question. In our experiments, we fine-tune SEAM-defended models under several realistic scenarios. When the model is fine-tuned on benign data, or on benign data that contains a small fraction of harmful samples, we do not observe collapse or noticeable degradation in utility (e.g., Tables 1 and 12). Similarly, fine-tuning on sensitive but non-harmful data also preserves model performance (e.g., Table 4). Collapse only appears under explicitly adversarial fine-tuning where the training set is dominated by harmful data. This matches our threat model, in which SEAM is designed to resist harmful fine-tuning rather than ordinary domain adaptation.
>
> From the provider perspective, SEAM introduces the familiar trade-off between utility and security that already exists in standard alignment. We expect that strong defenses such as SEAM will be most attractive in high-risk applications, for example, legal or medical assistants, where preventing misuse is more important than supporting arbitrary downstream fine-tuning.
>
> ---
>
>
> > The first sentence seems to have a wrong \citep command which does not render the citations properly.
>
> We thank the reviewer for pointing this out. We have fixed the citation.
>
> ---
>
> > Would it be possible to undo the SEAM training by training to maximize the self-destruction loss (essentially training with negative beta)?
>
> To address this question, we implement a reversal attack that augments the adversary's loss function with the $L_{\text{sd}}$​ loss term (using $\beta = -0.01$), thereby simultaneously pursuing the adversarial objective while attempting to escape the gradient trap (see Appendix C.5 for details).
>
> The table below reports post-attack harmfulness scores (HS) and zero-shot scores (ZS) across different learning rates $\eta$, showing that SEAM remains robust against this reversal attack. We attribute this resilience to the difficulty of retracing the optimization trajectory once the model has converged to a gradient-trap basin: the highly non-convex loss landscape makes it challenging to reverse the path by simply maximizing $\mathcal{L}_{\text{sd}}$​ from that local region.
>
> | $\eta$ = 2e-5 |      | 5e-5 |      | 8e-5 |      | 1e-4 |      | 2e-4 |      |
> |:--------:|:----:|:----:|:----:|:----:|:----:|:----:|:----:|:----:|:----:|
> |    HS    |  ZS  |  HS  |  ZS  |  HS  |  ZS  |  HS  |  ZS  |  HS  |  ZS  |
> |    5.3   | 49.3 |  8.2 | 35.5 |  0.0 | 25.6 |  0.0 | 25.1 |  0.0 | 26.2 |
>
> ---
>
> > Could SEAM trigger self-destruction when finetuned on some out-of-distribution domain-specific datasets that might come around as potentially not safe? I am thinking of medical datasets where medical advice could be classified by some models as harmful, or science datasets which could contain descriptions of dangerous chemicals / details about how atomic bombs work etc.
>
> Following the reviewer’s suggestion, we fine-tune SEAM-defended models using the OR-Bench benchmark [1], which consists of sensitive but not harmful data. The results indicate that self-destruction effect will not be triggered under that scenario (more details in Section 5.4 of the revised paper).

---

> ### Author Response · Authors · 2025-11-20
>
> > Is it possible that SEAM-trained models might be vulnerable to poison attacks? I think that including a few harmful (or even specifically crafted) examples in a benign dataset could cause the model to collapse. Would it be possible to actually identify such samples from their gradient information?
>
> We conduct additional experiments that mix harmful data into benign data. We construct mixed datasets by combining varying proportions of harmful data (BeaverTails) with clean data (Alpaca), then evaluate both harmfulness and utility under Attack #2 from Figure 4(a) on llama2-7b. The table below summarizes the post-attack harmfulness score (HS) and zero-shot score (ZS) for different harmful data contamination ratios $p$. Notably, SEAM shows graceful utility degradation that scales with the contamination level.
>
> |           | Pre-attack |      | $p$ = 0.0 |      | 0.01 |      | 0.05 |      | 0.10 |      | 0.20 |      |
> |-----------|------------|------|---------|------|------|------|------|------|------|------|------|------|
> |           | HS         | ZS   | HS      | ZS   | HS   | ZS   | HS   | ZS   | HS   | ZS   | HS   | ZS   |
> | Base      | 5.0        | 51.6 | 5.0     | 51.2 | 12.6 | 51.2 | 27.5 | 53.2 | 50.9 | 51.9 | 76.6 | 51.5 |
> | SEAM      | 3.8        | 50.9 | 4.0     | 51.5 | 5.5  | 51.2 | 5.5  | 52.6 | 9.1  | 50.7 | 9.5  | 50.8 |
>
> Regarding using gradient information to detect harmful samples, we identified a recent work [4] that leverages gradient signals to identify harmful samples and perform "surgery" to purify them, demonstrating the potential of using gradient information for defensive purposes.
>
>
> ---
>
> [4] Yi et al., SafeGrad: Gradient Surgery for Safe LLM Fine-Tuning, 2025
>
> ---
>
>
> Again, we thank the reviewer for the valuable feedback. Please let us know if there are any other questions or suggestions.
>
>
> Best,
>
> Authors

---

> > ### Comment · Reviewer_Y52o · 2025-11-24
> >
> > Thank you for the detailed rebuttal! The additional experimental results have answered my questions. I like the choice of fine-tuning on the OR-bench dataset as these samples are purposefully designed to be similar to malicious prompts. I will change my recommendation to 8: Accept.
> >
> > PS: I still think readers might be interested in some experiments with fine-tuning and evaluating SEAM-trained models on domain-specific datasets (e.g. medical, math, psychology etc.) and comparisons with the base models. Showing that SEAM models can achieve good performance on such datasets after fine-tuning could further strengthen the **camera-ready** paper. While the OR-Bench fine-tuning experiment convinced me from a theoretical perspective, it does not represent a very practical fine-tuning application.

---

> ### Author Response · Authors · 2025-11-27
>
> Dear Reviewer Y52o,
>
> We are delighted that our response addresses your questions and sincerely appreciate your willingness to consider raising the score to 8!
>
> Following your suggestion, we further evaluate the trainability of SEAM using domain-specific subsets of the MMLU dataset [1]. MMLU is a multiple-choice question answering benchmark that covers 57 diverse subjects. We group selected subjects into three broad representative domains: Math\&Logic, Medical, and Law\&Politics (etails in Appendix C.12 in the revised manuscript).
>
> Table below presents the fine-tuning score (FS) comparison between SEAM and the base model after fine-tuning on data from the corresponding domains. Specifically, we use the test' split of the dataset in the domain for training and measure choice accuracy on the corresponding validation' split as FS, where the answer choice is extracted from the model's response by GPT-4o-mini. As shown in the table below, SEAM achieves FS fairly close to the base model, indicating that the self-destructive property has minimal impact on trainability across various domains.
>
>
> | Model\Domain | Math\&Logic |  Medical | Law\&Politics |
> |--------------|:----:|:----:|:----:|
> | Base         | 44.7 | 51.2 | 52.8 |
> | SEAM         | 43.3 | 51.8 | 53.5 |
>
>
> [1] Hendrycks et al., Measuring Massive Multitask Language Understanding, ICLR’21
>
> ---
>
>
> Again, we sincerely appreciate your feedback. Please let us know if you have any further questions or suggestions.
>
> Best,
>
> Authors

---

### Author Response · Authors · 2025-12-02
**Rebuttal Summary**

Dear AC, SAC, and PCs,

Thank you for your time and effort in reviewing our paper. To provide a comprehensive view of our work and responses, we include below (1) an overview of our paper, (2) a summary of reviewers’ comments and updates, and (3) a brief walkthrough of the rebuttal process.

---

**[Overview]**

We introduce SEAM, a new defense paradigm that makes LLMs inherently resistant to harmful fine-tuning. SEAM couples benign and harmful optimization trajectories, using adversarial gradient ascent to induce self-destructive collapse under malicious training. We enable this approach with an efficient Hessian-free estimator that preserves benign utility while achieving robustness against varying harmful fine-tuning attacks.

---

**[Reviewers’ Comments and Updates]**

- Reviewer Y52o is positive about our work (with an initial score of 6), with primary questions regarding the effects of benign fine-tuning and adaptive attacks. **He indicated that he will raise the score to 8 based on our response.**

- Reviewer tToc is supportive of our work (with an initial score of 6), with main questions concerning adaptive attacks and threat model clarification.

- Reviewer b8Na is positive about our contributions (with an initial score of 6), with primary questions regarding more sophisticated adaptive attacks and threat model clarification.

- Reviewer 1ay1 is highly positive (with an initial score of 8), with main questions about the effects of benign fine-tuning and adaptive attacks.

---

**[Rebuttal Walkthrough]**

We have provided point-by-point responses to all reviewers' questions (in detailed responses), with all revisions highlighted in the revised manuscript. Below is a summary of the major questions we have addressed.

**1. Robustness under adaptive harmful-data attacks. (Reviewers Y52o, tToc, b8Na, 1ay1)**

The reviewers raised a wide range of adaptive harmful fine-tuning attacks for stress-testing. We added a wide range of adaptive attack evaluations, including low-learning-rate extended attacks (up to 50K steps), reversal attacks with negative self-destructive loss, full/partial LoRA module attacks, layer-freezing strategies, poisoning-ratio sweeps (1–20%), and weight-orthogonalization attacks (Appendix C.5). These experiments consistently show that SEAM remains robust across diverse adversarial settings.

**2. Preservation of utility and harmlessness under benign fine-tuning (Reviewers Y52o and 1ay1)**

The reviewers questioned the behavior of benign fine-tuning. We added extensive experiments using ORBench, XSTest,  and domain-specific MMLU subsets (Math&Logic, Medical, Law&Politics). Results show SEAM maintains harmlessness, low incorrect-refusal rates, and comparable zero-shot performance to base models across these benign scenarios (Section 5.4 and Appendix C.12).

**3. Threat model and practical scenario clarification (Reviewers Y52o, tToc, b8Na)**

We clarified that SEAM is intended for high-risk deployments involving moderately sized open-source models, where white-box harmful fine-tuning is the primary threat. Under this setting, SEAM preserves utility during benign fine-tuning and only degrades when the training data is dominated by harmful content, consistent with the threat model. We also demonstrate in Table 7 that SEAM’s configuration generalizes well across a wide range of open-source models.

---

**[Final Remark]**

We believe our rebuttal and additional experiments have thoroughly addressed the reviewers' questions. Please let us know if you have any further questions or suggestions.

Best regards,

 Authors

---

### Meta-Review · Area_Chair_9G6Z · 2026-01-06

**Summary:**

The authors provide an alignment technique that aims at defending against fine-tuning attacks. In particular, the proposed technique induces a self-destructive effect on the model's weight when finetuned on harmful data, making that finetuned model unusable.

The reviews have been overwelmingly positive but none of them have been outstanding, I thus recommend this paper for a poster.

**Reviewer Concerns:**

The concerns have been addressed.

**Reviewer Scores:**

The only reviewer who responded to the rebuttal said they would increase their score. I believe that the other reviewers would have maintained their positive score after the discussion.

---

### Decision · Program_Chairs · 2026-01-26

Accept (Poster)